# Self-Assembly and Magnetic Order of Bi-Molecular 2D Spin Lattices of M(II,III) Phthalocyanines on Au(111)

**Miloš Baljozović** [1,*], **Xunshan Liu** [2], **Olha Popova** [3], **Jan Girovsky** [1], **Jan Nowakowski** [1], **Harald Rossmann** [1], **Thomas Nijs** [3], **Mina Moradi** [1,4], **S. Fatemeh Mousavi** [3], **Nicholas C. Plumb** [5], **Milan Radović** [5], **Nirmalya Ballav** [6], **Jan Dreiser** [5], **Silvio Decurtins** [2], **Igor A. Pašti** [7,8], **Natalia V. Skorodumova** [8,9], **Shi-Xia Liu** [2,*] and **Thomas A. Jung** [1,3,*]

[1] Laboratory for Micro- and Nanotechnology, Paul Scherrer Institute, 5232 Villigen-PSI, Switzerland; vedatorsk@gmail.com (J.G.); jan.andrzej.nowakowski@gmail.com (J.N.); h_rossmann@gmx.at (H.R.); minamoradi.1989@gmail.com (M.M.)

[2] Department of Chemistry, Biochemistry and Pharmaceutical Sciences, University of Bern, Freiestrasse 3, 3012 Bern, Switzerland; xliu350@zstu.edu.cn (X.L.); Silvio.decurtins@unibe.ch (S.D.)

[3] Department of Physics, University of Basel, Klingelbergstrasse 82, 4056 Basel, Switzerland; ollia.popova@gmail.com (O.P.); thomask.nijs@gmail.com (T.N.); fatemeh.mousavi@unibas.ch (S.F.M.)

[4] School of Life Science, University of Applied Sciences and Arts, Northwestern Switzerland, Gründenstrasse 40, 4132 Muttenz, Switzerland

[5] Swiss Light Source, Paul Scherrer Institute, 5232 Villigen-PSI, Switzerland; nicholas.plumb@psi.ch (N.C.P.); milan.radovic@psi.ch (M.R.); jan.dreiser@psi.ch (J.D.)

[6] Department of Chemistry, Indian Institute of Science Education and Research, Pune, Maharashtra 411008, India; nballav@iiserpune.ac.in

[7] Faculty of Physical Chemistry, University of Belgrade, Studentski trg 12-16, 11000 Belgrade, Serbia; igor@ffh.bg.ac.rs

[8] Department of Materials Science and Engineering, School of Industrial Engineering and Management, KTH—Royal Institute of Technology, Brinellvägen 23, 100 44 Stockholm, Sweden; snv123@kth.se

[9] Department of Physics and Astronomy, Uppsala University, Box 516, 751 20 Uppsala, Sweden

[*] Correspondence: milos.baljozovic@psi.ch (M.B.); shi-xia.liu@unibe.ch (S.-X.L.); thomas.jung@psi.ch (T.A.J.)

**Abstract:** Single layer low-dimensional materials are presently of emerging interest, including in the context of magnetism. In the present report, on-surface supramolecular architecturing was further developed and employed to create surface supported two-dimensional binary spin arrays on atomically clean non-magnetic Au(111). By chemical programming of the modules, different checkerboards were produced combining phthalocyanines containing metals of different oxidation and spin states, diamagnetic zinc, and a metal-free 'spacer'. In an in-depth, spectro-microscopy and theoretical account, we correlate the structure and the magnetic properties of these tunable systems and discuss the emergence of 2D Kondo magnetism from the spin-bearing components and via the physico-chemical bonding to the underlying substrate. The contributions of the individual elements, as well as the role of the electronic surface state in the bottom substrate, are discussed, also looking towards further in-depth investigations.

**Keywords:** self-assembly; metal phthalocyanines; scanning tunneling microscopy (STM); X-ray absorption spectroscopy (XAS); X-ray magnetic circular dichroism (XMCD); Kondo screening; Ruderman–Kittel–Kasuya–Yosida (RKKY) exchange interaction

## 1. Introduction

In the fields of molecular electronics and spintronics, and even more so for the fabrication of functional architectures for quantum technologies, it is vital to create atomically precise quantum architectures in an addressable manner. Aligning spin bearing molecules on surfaces and interfaces allows us to gain control of the magnitude and direction of their magnetic moments in a low-dimensional arrangement at the so-called 'spinterfaces' [1–4].

While the interest in the latter stems from the material-specific tuning of interface characteristics such as the spin-injection behavior, more recently 'true' single-layer two-dimensional (2D) quantum spin lattices have been prepared using on-surface supramolecular chemistry and/or surface nanostructuring with different, including frustrated, exchange interactions [5,6]. Such spin architectures are of high interest for fundamental research into low-dimensional magnetic materials and are also very compelling for research into future quantum technologies. This is because the coupling between isolated spin systems can be mediated via their coordination architecture and/or via their interaction with the magnetic or non-magnetic bottom substrate [7–11]. The physical properties of these complex supramolecular architectures can be defined by the spin bearing ion or coordination cluster; their hosting molecular platform e.g., phthalocyanines and porphyrins or calixarenes, among other suitable compounds; their chemical functionalization and directed self-assembly into well-defined arrays. Remarkably, the magnetic properties of these single-layer spin systems are generally accessible by spectro-microscopy correlation, combining element sensitive X-ray photoelectron spectroscopy (XPS) and magnetic moment sensitive X-ray magnetic circular dichroism (XMCD) with local probe microscopy such as scanning tunneling microscopy (STM). The latter technique also identifies the presence or absence of magnetic states in spin-dependent tunneling experiments or from local density of states (DOS) measurements, revealing the Kondo shielding of magnetic moments, as well as the presence or absence of chemical ligands modifying the spin-bearing ions [12–26].

Towards these ends, metal-porphyrins and phthalocyanines have provided very robust and versatile platforms for embedding specific magnetic ions selected from a wide range of elements to modify their coordination shell, their electronic and magnetic state, and also to direct their on-surface architecturing via selected functional units [27–39]. In this way, precise supramolecular on-surface architectures have been built, e.g., by the use of fluorine-substitution, by site recognition on patterned substrates, and by their integration into surface coordination polymers and network architectures. Here, non-covalent interactions such as hydrogen bonding, metal-organic coordination, van der Waals, or ionic interactions, perform this role [40–45].

With regard to low-dimensional spin-architectures, complex supramolecular materials have been developed and investigated for their interesting properties, e.g., by site-specific ligation [10,46–48]. Well-ordered two-component structures of CoPc and CuTPP (TTP = tetraphenylporphine) on an Au(100) substrate showed alternating, hence segregated, single component chains [49]. Similarly, a supramolecular checkerboard structure was described by Yoshimoto et al. for a ZnPc and octaethylporphyrin ZnOEP system deposited on Au(111) and Au(100), whereby the study also emphasized the role of the atomic arrangement of the underlying substrate [50]. Moreover, Sabik et al. studied CoPc + CuF$_{16}$Pc arrangements on Ag(100) for selected ratios between the two molecular components [51–53]. Note that bi-component checkerboards also emerged by the site-specific adsorption of non-functionalized porphyrins on a reconstructed Bi on Cu(100) substrate [54].

Wäckerlin et al. investigated a 2D electron spin array with a checkerboard pattern by co-deposition of MnPc and fluorinated FeF$_{16}$Pc on an oxygen-reconstructed ferromagnetic Co substrate. Thereby, the coupling of the magnetic moments of the spins of the metal ions with the magnetic substrate could be probed, as well as the function of the axial ligand coordination [55]. Interestingly, depositing the same MnPc–FeF$_{16}$Pc system on Au(111) leads to an analogous ordered surface pattern, whereby long-range ferrimagnetic ordering in the monolayer 2D Kondo lattice was observed [5]. In this system, the interaction of the molecular spin centers with the substrate conduction electrons gives rise to remanent long-range magnetism in the 2D layer. The magnetic moments in this system interact with the conduction electrons in the underlying substrate and the order is attributed to the Ruderman–Kittel–Kasuya–Yosida (RKKY) oscillatory interaction, in spite of the Kondo screening partially reducing the effective magnetic moments. It is worth noting that this observation contrasts with the Mermin–Wagner theorem for 2D systems with the Heisenberg model under the assumption of isotropic magnetic coupling [56]. It is

the particular symmetry-breaking role of the ad-layer with the substrate that breaches this assumption. The spin centers significantly interact, in particular with the surface electronic states, as is also evidenced by a surprisingly strong out-of-plane component in the remanent magnetization of the 2D symmetric assembly of planar molecules.

In the present study we systematically modified this supra-molecular checkerboard architecture to investigate the influence of the chemical state, the d-state population, and the spin of the center metal atoms on the long-range magnetic ordering induced by the exchange interaction with and via surface electrons (Scheme 1, bottom). Chemically modified 2D spin-architectures of phthalocyanines and fluoro-functionalized phthalocyanines equipped with different metal centers (Fe, Mn, MnCl Co, Cu, Zn, and metal free, see Scheme 1, top) were created according to different recipes on Au(111). Zinc (Zn) was chosen for its diamagnetic behavior in the 3D crystalline (bulk) form. An important motivation for this selection of molecular modules was provided by their different filling of the 3d-orbitals at the central metal atom that is reflected in their different spin ground states [57,58]. The details of the coordination sphere and the spin state govern the interaction with the substrate [57,59] and are also expected to affect the magnetic exchange coupling via the surface atoms and surface states, which is the topic of our investigation and report.

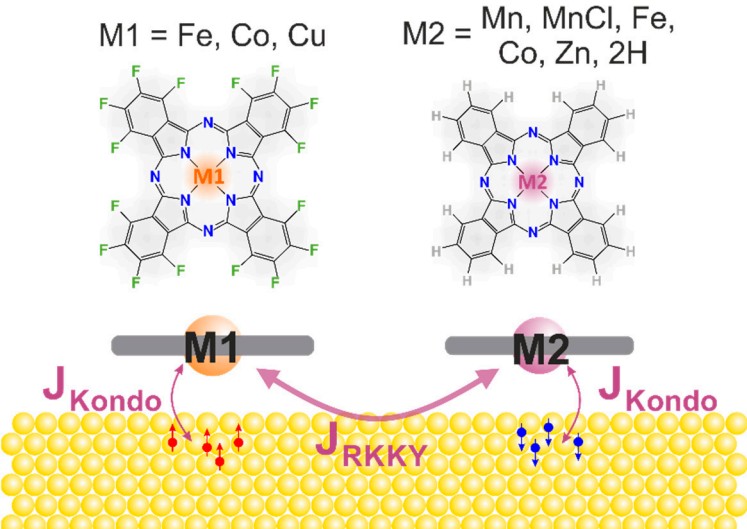

**Scheme 1.** Molecular structures (**top**) and schematic representation of 2D checkerboard assembly of phthalocyanine molecules on an Au(111) substrate, together with the involved competing interactions, namely RKKY and Kondo interactions (**bottom**).

## 2. Results and Discussion

In the first step of the aforementioned investigations, the procedure for preparation of alternating assemblies of two phthalocyanine molecules was adapted in line with Sabik et al. [51–53]. We departed from our previously established procedure, where two molecular species, namely MnPc and FeFPc were co-evaporated onto the substrate of choice, kept at room temperature [5,55]. In the present work, in contrast, the evaporation rate of each molecule was tuned independently using a QCMB, and the molecules were subsequently deposited on Au(111). The amount of each molecular species was then monitored by XPS, looking into the ratios of C1s, N1s, and F1s photoelectron peaks, and adjusted if necessary. It was an important first result of our present study that subsequent molecular deposition leads to the same compositionally ordered checkerboard-like structure on Au(111) as in the case of the co-evaporation of two molecular species.

Figure 1 contains two STM images obtained at 4.5 K upon subsequent (Figure 1a) or simultaneous (Figure 1b) deposition of MnPc and FeFPc molecules onto Au(111) crystal kept at room temperature. In both cases, the distinct appearance of the two molecules

confirmed the checkerboard-like structure. Interestingly, at the terraces with lower local molecular coverage next to the extended islands of co-assembled molecules, chains of alternating MnPc and FeFPc molecules along the herringbone reconstruction of the Au(111) substrate are observed (Figure S1). This behavior corresponds to the different surface structures and surface charging of the fcc, hcp, and 'ridge' regions of the top atomic layers in this complex surface reconstruction [60]. The 2D intermolecular binding motif overrules this site-specificity as the coverage increases. Note that it comprises a practical advantage, in the present work and beyond, that the amount/ratio of molecules can be corrected by adding the missing component in a subsequent deposition step until the XPS signals reach the proper ratio. In contrast, the alternative co-evaporation procedure often leads to unsatisfactory coverage and stoichiometry of the two components, unsuitable for XAS/XMCD investigations.

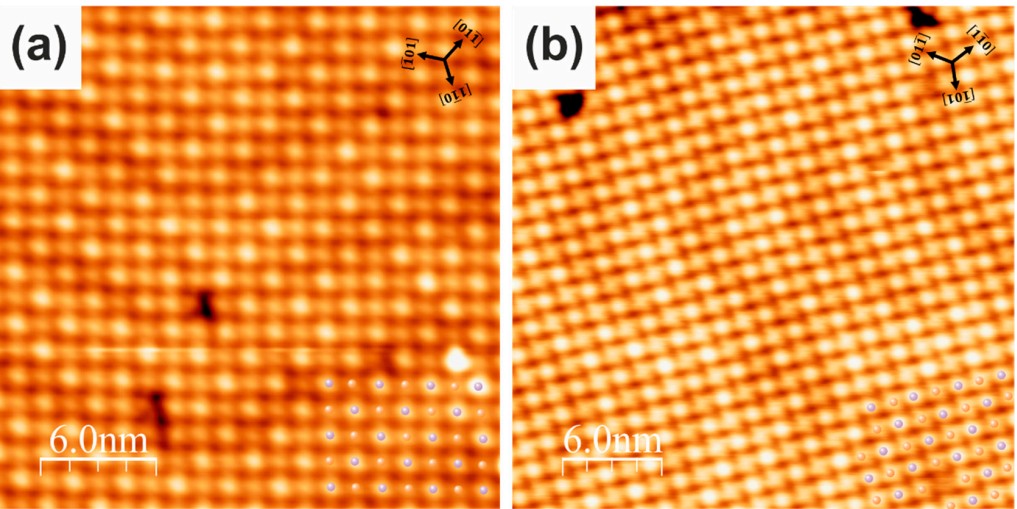

**Figure 1.** STM micrographs of the checkerboard assembly of MnPc and FeFPc molecules on Au(111). (**a**) 35 nm × 35 nm micrograph (U = 0.1 V, I = 5 pA) obtained upon subsequent deposition of the molecules. (**b**) 30 nm × 30 nm micrograph (U = −2.2 V, I = 5 pA) obtained upon co-deposition of the molecules. Irrespective of the sequence of deposition steps we obtained the same checkerboard morphology with herringbone reconstruction still present. Colored circles in the bottom right corner of both images serve as a guide for the eye that illustrates the checkerboard pattern of Fe (orange) and Mn (violet) in the assembly.

Following the successful STM investigation of subsequently deposited MnPc and FeFPc molecules arranged in a checkerboard manner (Figure 1a,b), the magnetic properties of this system were investigated by means of XAS/XMCD. The samples were probed at 2.5 K upon cooling from room temperature and in normal X-ray incidence, as indicated in the sketch of the experimental set-up (Figure 2a). XAS (black) and XMCD $L_{2,3}$ spectra of Fe and Mn acquired at 0.0 T (red), +6.8 T (green), and +0.0 T (blue) are shown in Figure 2c,d. The plus sign stands for the fact that the magnetic field was ramped down from a positive, non-zero value. The illustrations shown in Figure 2b serve as guidance of the evolution of magnetic properties of this system and the formation of 2D-ferrimagnetic ordering, as followed by XAS/XMCD: Initially, upon cooling the sample to 2.5 K and while keeping it at 0.0 T, there is no long-range order present, and spins of Mn and Fe are randomly oriented "up" and "down". Spatially averaging techniques such as XAS/XMCD thus display no apparent magnetic signal (Figure 2c,d red). This is supported by the high out-of-plane orbital anisotropy reported for the MnPc+FeFPc/Au(111) system at low temperatures [5], which does not allow spontaneous long-range ordering. In a high magnetic field, namely at 6.8 T applied in a direction perpendicular to the sample surface, the anisotropy energy, as well as thermal spin fluctuations, are overruled, and the spins of both Mn and Fe

become aligned along the magnetic field direction (Figure 2c,d green). At this high applied magnetic field, the XAS/XMCD shapes and intensities, as well as the dominant orbital component of Mn and Fe spins, were in very good agreement with the earlier report where the samples had been prepared by co-evaporation [5].

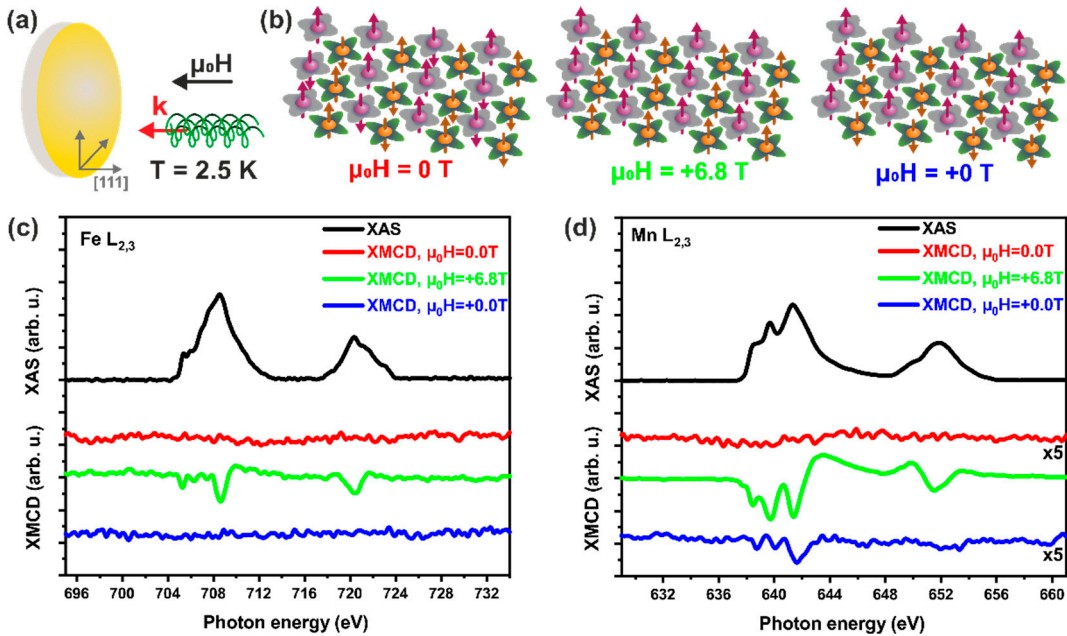

**Figure 2.** XMCD investigation of the magnetic properties of the binary MnPc+FeFPc/Au(111) system. (**a**) Sketch of the experimental set-up of the XAS/XMCD measurements in normal X-ray incidence: the external magnetic field and the k-vector of the X-rays are parallel to the surface normal along the crystallographic [111] direction. (**b**) Illustration of the initially disordered molecular spins at 0.0 T, the high field enforced, the parallel (ferromagnet-like) aligned spins at +6.8 T and the ferrimagnetically ordered spins at +0.0 T after initial magnetization. XAS (black) and XMCD $L_{2,3}$ spectra of Fe (**c**) and Mn (**d**) in chronological order acquired in normal X-ray incidence at 0.0 T (red), +6.8 T (green), and +0.0 T (blue). The XAS is plotted upon background subtraction. The XMCD spectra of Mn at 0.0 T (red) and +0.0 T (blue) are enhanced by a factor of 5 for better visibility. No XMCD was detected in the initial measurement at 0.0 T; it only appeared upon magnetization at +6.8 T. All experiments were performed at 2.5 K.

Upon the initial magnetization in a high magnetic field, the Fe and Mn moments were found to be in an antiparallel orientation after switching off the magnetic field (Figure 2c,d blue). This is in accordance with our previous report where such remanent magnetization was observed up to ~3.7 K and was attributed to the Ruderman–Kittel–Kasuya–Yosida (RKKY) oscillatory interaction via the surface electrons of Au(111) in conjunction with partial Kondo shielding [5]. In both cases, Mn spins, being in the higher spin ground state due to their higher number of unpaired electrons compared to Fe, retain the magnetic orientations imposed by the high magnetic field. The Fe spins, in contrast, collectively revert to the opposite direction, driven by the exchange interaction.

Mechanistically, this collective behavior can be understood by accounting for the preferred out-of-plane anisotropy of the Fe and Mn spins on Au(111) at 2.5 K, as well as for the distance-dependent oscillatory behavior of the RKKY coupling [5,61,62]. The nearest neighbor distance between spin bearing centers of the same kind (Mn-Mn or Fe-Fe) thus falls into the ferromagnetic coupling regime. In contrast, the distance between the Mn and Fe spin centers, i.e., the separation between the two sublattices, falls into the antiferromagnetic coupling regime of the spatial oscillation of the RKKY interaction. The spatial frequency is determined by the Fermi wave vectors $k_F$ of the Au(111) surface state (Figure S2 top). This, in turn, means that all Mn, and respectively Fe, spin in the supramolecular assembly couple ferromagnetically via RKKY to their neighbors of the same kind and form sublat-

tices of opposite magnetization, inducing a long-range antiferromagnetic order (Figure S2 bottom). It is exactly this long-range ordering mechanism via the delocalized conduction electrons in the substrate that induces the collective behavior of the aforementioned spins. As the strong magnetic field is reduced/switched off, the Fe spin direction is reversed by the RKKY interaction, leading to the formation of an extended 2D ferrimagnet. The switching magnetic field corresponds to the RKKY exchange coupling strength between the different magnetic ions. This particular collective behavior emphasizes the importance of high magnetic fields for ordering the magnetic moments in the present 2D spin-arrays. The high magnetic field may be essential for orienting the spins in such supra-molecular architectures, due to the limited (2D) dimensionality and the rather weak RKKY interaction. Spontaneous ordering at zero field may be limited to very small domains or some degree of nearest/next nearest neighbors very different from the well-known cases of conventional bulk magnetic materials.

In order to investigate the modifications of the Au(111) surface states accompanying the formation of the aforementioned supramolecular 2D lattices on top of the Au(111) substrate, we employed angle resolved photoelectron spectroscopy (ARPES). Particularly, we intended to investigate how the formation of different layers containing molecules with and without spin centers influences the surface states of Au(111). This can be done by looking at the shape of the Fermi surface in proximity to the interface and in particular at the sections of the Fermi surface along the kx and ky directions. In order to do so, several 2D systems with increasing complexity (clean Au(111), 2HPc/Au(111), 2HPc+FeFPc/Au(111), and MnPc+FeFPc/Au(111)) were investigated (Figure S3). Intriguingly, the spin texture was not observed to depend on the layer architecture, most likely due to the experimental constraint of a not sufficiently low sample temperature. The determined surface state shifts towards the Fermi level and the spectral broadening, caused by the Pauli repulsion between the surface state electrons and the adsorbate layer, are well in line with earlier reports [63–73]. More details can be found in the Supplementary Material.

Following an in-depth analysis of the 2D checkerboard layer containing MnPc and FeFPc molecules, the investigation was expanded to several other binary 2D-checkerboard systems on Au(111) containing phthalocyanine molecules. Fe, Co, and Cu containing perfluorinated phthalocyanine molecules (FeFPc, CoFPc, and CuFPc) on one side and Mn, MnCl, Fe, Co, Zn, and metal-free phthalocyanine molecules (MnPc, MnPcCl, FePc, CoPc, ZnPc, and 2HPc) on the other side were combined into seven pairs of molecules in an analogous manner to the FeFPc and MnPc pair. The magnetic properties of these layers, in particular the presence or absence of remanent magnetization, was then investigated by means of XAS/XMCD at 2.5 K. A summary of these investigations is presented in tabular manner in Figure 3a, while a complete XAS/XMCD dataset is shown in Figure S4. Surprisingly, the remanent magnetic moments, after switching off the magnetic field, were only found to be of non-zero value for the Fe/Mn pairing; the case already discussed above. This is an intriguing result, as the inter-molecular distances are governed by the FPc/Pc molecular frame and are expected to be the same for all the investigated systems. The molecular order was consistently confirmed by our STM data and was also present when the metal free (2HPc) was used (Figure S5). A similar or the same intermolecular spacing, in turn, should provide a prerequisite for the same exchange coupling mediated by the RKKY interaction, in particular if Fe or Mn is involved. In the cases where Fe or Mn containing phthalocyanines were combined with spin-inactive/absent ones, such as CoPc, ZnPc, or 2HPc we would expect that a ferromagnetic RKKY coupling of the Fe or Mn sublattices would be detected by the XMCD in remanence, i.e., after the initial magnetization at high field. Note here, that CoPc is known to be quenched on Au(111) due to valence fluctuations [74] and shows no XMCD, even at +6.8 T (Figure S4b,e,f). ZnPc (diamagnetic, with a fully filled d-shell) or 2HPc provide similar cases. The absence of a detectable remanence was probably linked to the expected, lower critical temperatures of these molecule pairings; due to the decay of the RKKY interaction with distance, the ferromagnetic exchange interaction between the next-nearest neighbors in these binary

checkerboards is weaker than the anti-ferromagnetic interaction between the nearest neighbors. In the absence of a one spin-sublattice the magnetic stabilization/ordering of these spin layers may not be strong enough for a magnetic long-range order to occur up to or above the lower limit of ~2.5 K to perform XMCD experiments with the setup used.

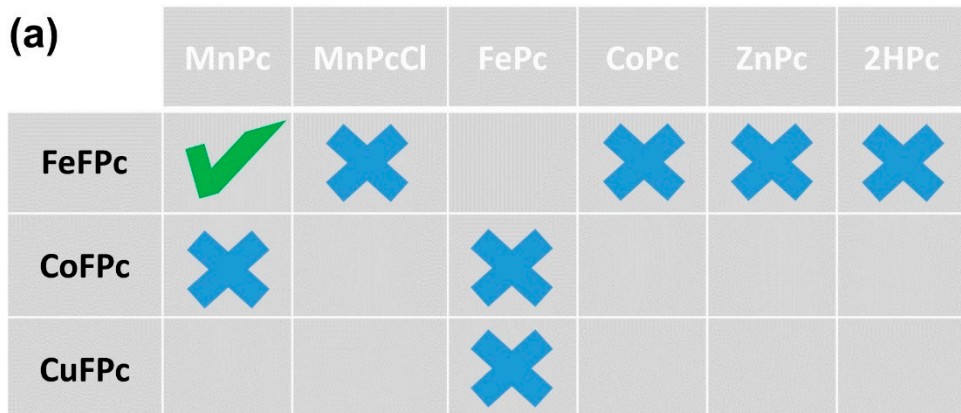

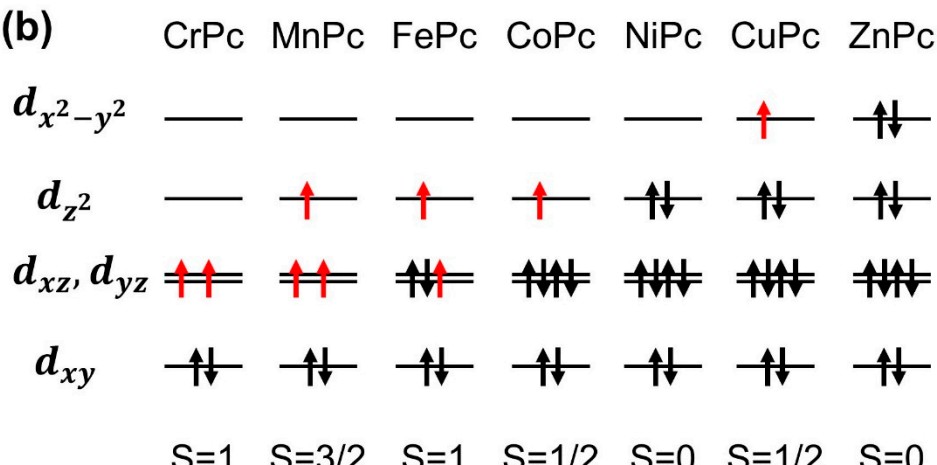

**Figure 3.** Overview of the investigated binary assemblies and electronic structures of individual phthalocyanine molecules. (**a**) Tabular representation of the investigated binary systems and summary of our extensive XAS/XMCD investigations: at 2.5 K, magnetic remanence of the 2D spin-checkerboard was observed exclusively for MnPc and FeFPc. (**b**) Electron filling scheme for the different MPcs: (black) indicates paired and (red) indicates un-paired electrons in $D_{4h}$ symmetry. Below, the total spin due to the unpaired electrons has been listed. Note that this configuration is prone to change, as the molecules experience a modified ligand field ($C_{4v}$), e.g., by their adsorption on the surface or by ligand interaction/coordination.

For a discussion about the absence of remanence in the other investigated systems, one has to account for the spin states of these molecules. The filling of the outer shell of the 3d metal containing phthalocyanine molecules, as well as their accompanied spin states, are listed in Figure 3b for comparison. Among the investigated systems, we found values in the range from spin 0 to spin 3/2 between, e.g., Zn and Mn. Among them, however, only Mn and Fe possessed a spin state higher than 1/2. It is important to note that the Kondo-shielding of the adsorbed spin-state is induced by the substrate conduction electrons and works against the simultaneous action of the RKKY interaction inducing the order [5]. In case of perfect Kondo screening of, for example, an impurity spin system surrounded by/closely interacting with conduction electrons, the RKKY interaction would most likely not occur. The higher spin value allows for higher magnetic moments that are expected to induce a less perfect Kondo shielding in an on-surface adsorption geometry.

Therefore, the checkerboard of Mn and Fe phthalocyanines was expected to show the strongest effect. The other investigated metal pairings, might similarly be expected to order at critical temperatures below the present 2.5 K limit. In this light, lower experimental temperatures become essential to further investigate the phenomenon, while CrPc with its different electronic shell configuration might be another interesting candidate for future experiments.

An interesting case to discuss is the absence of remanence in the case when FeFPc and MnPcCl were combined in a checkerboard assembly. By adding the axial ligand to the Mn, oxidation state of Mn changes from 2+ to 3+, associated with a high spin compound with S = 2 [75]. This transition was further corroborated by the shape and the strongly isotropic magnetic signal in our XAS/XMCD measurements with the applied high magnetic field (Figure S4a). Consistent with the significant orbital moments in our XMCD data, the axial and the surface ligand together bring the ligand field closer to octahedral geometry [10,76]. Thus, the $d_{z^2}$ orbital of Mn is lifted in energy and remains empty. This electron arrangement is also responsible for the almost complete reduction of the magnetic anisotropy observed in XAS/XMCD experiments. Due to this very high spin state, which is maintained in the case of surface adsorption, we can exclude complete Kondo screening as a reason for the absence of RKKY interaction. From the high magnitude of the magnetization, in contrast, a higher critical ordering temperature via the RKKY interaction could be expected in comparison to that of the MnPc + FeFPc pair. The lack of ordering at 2.5 K in this case, as was observed in our data, reflects the pivotal role of the $d_{z^2}$ orbital and its occupancy for sustaining the RKKY interaction. Such an interpretation is intuitive, because the $d_{z^2}$ orbital, indeed, has a favorable symmetry that facilitates the interaction with the underlying surface and in particular the surface state electrons necessary for the RKKY interaction. This is also in line with the Mermin–Wagner theorem for 2D systems with the Heisenberg model, predicting that the absence of magnetic anisotropy or symmetry breaking will prevent magnetic ordering at finite temperatures. Chlorine as an axial ligand acts towards restoring the symmetry of the 2d layer as it had been previously broken by the interaction with the bottom substrate. At the same time it lifts the magnetic anisotropy by depopulation of the highly directional $d_{z^2}$ orbital. A further, deepened understanding of the mechanisms of RKKY interaction in spin-bearing and surface supported molecular architectures opens up new methods for the design and manipulation of their magnetic properties, either by chemical modification of the components, i.e., here phthalocyanine molecules, or by use of external chemical stimuli.

In order to glimpse into and better understand the role of d-orbital filling for the interaction with the underlying substrate, a necessary component for retaining the RKKY interaction and 2D-ordering, we performed numerical simulations. The DFT+U method including dispersion correction, was employed. Individual molecules (Mn-CuPc) were placed on top of a three-layer Au(111) slab and optimized. It has already been shown that the interaction of Mn-CuPc molecules with Au(111) is rather weak, with almost no structural changes of the underlying substrate, and that different configurations of MPc molecules on the substrate differ to a very small extent [5,57]. Girovsky et al. [5] reported an average adsorption energy of −0.88 eV for co-adsorbed MnPc and FePc molecules on Au(111), with somewhat reduced magnetic moments of adsorbed molecules compared to the isolated ones. Zhang et al. [57], however, found that the adsorption energy decreases as the metal center changes with position in the periodic system, from Mn to Cu, and the magnetic moments of isolated molecules are preserved upon adsorption on Au(111), with the only exception being CoPc. This clearly indicates that the calculated magnetic properties of MPc@Au(111) are highly sensitive to the applied computational approach. Moreover, the adsorption energy was the highest for MnPc (−0.53 eV) and dropped to −0.33 eV for CuPc. We found that all the studied molecules interact with Au(111) via $d_{z^2}$, $d_{yz}$, and $d_{xz}$ orbitals, which were found to be delocalized and to overlap with the Au(111) d-states, which is apparent when compared to the states of isolated molecules (Figure S6). This orbital overlapping is more pronounced for MnPc and FePc, and to some extent CoPC,

than for NiPc and CuPc. In the latter case, the d-orbitals of M centers are highly occupied and centered at low energies, which is in line with the reduced interactions of MPc with Au(111) along the Mn-Cu series. These findings are also in line with our experimental observations of RKKY ordering for the Mn and Fe co-assembly.

We further considered the interaction of MnPc with Au(111), which was characterized by Mn protrusion from the Pc plane (the molecule is, otherwise, planar in the isolated state), in line with previous findings [5,57]. Note that a small protrusion of the metal center was also observed for FePc. This is an indication of the strong coupling between the Mn/Fe metals, between the molecules and the electronic states in the Au substrate, which is also clearly visible when the spin densities of the respective systems are considered (Figure S6). The interaction of the spin moment for MnPc with the Au substrate atoms, however, disappears (for the same spin density considered), when Cl is present at the Mn center; that is, when MnPcCl is adsorbed on Au(111) (Figure S7). To further investigate the origin of this effect, we analyzed the orbital projected densities of states of Mn and Cl atoms in isolated and adsorbed MnPc and MnPcCl molecules (Figure 4). The protrusion of Mn towards the off-surface direction of the Pc plane in adsorbed MnPcCl weakens the interaction with the substrate compared to adsorbed MnPc. In addition, we find here, that the Mn $d_{z^2}$ states are decoupled from the Au d-states. This is due to the interaction of the Mn $d_{z^2}$ states with the Cl $p_z$ states forming the Mn-Cl bond. The decoupling in turn killed the RKKY interaction in our simulations, confirming the pivotal role of the Mn $d_{z^2}$ states for long-range ordering via the surface electrons. The charge is transferred to Cl (0.62 e), and the Mn $d_{xy}$ orbitals responsible for Mn-N bonding shift above the Fermi level. This process also leads to the reduction of the total magnetic moment of the system by 0.92 $\mu_B$. The electron population on Cl is thereby increased and the Mn-Cl bond is 0.15 Å longer upon adsorption of MnPcCl compared to the isolated MnPcCl molecule. This is a consequence of a smaller charge transfer (0.52 less) within the isolated MnPcCl, not adsorbed on any interacting substrate, resulting in a smaller ionic radius of the Cl.

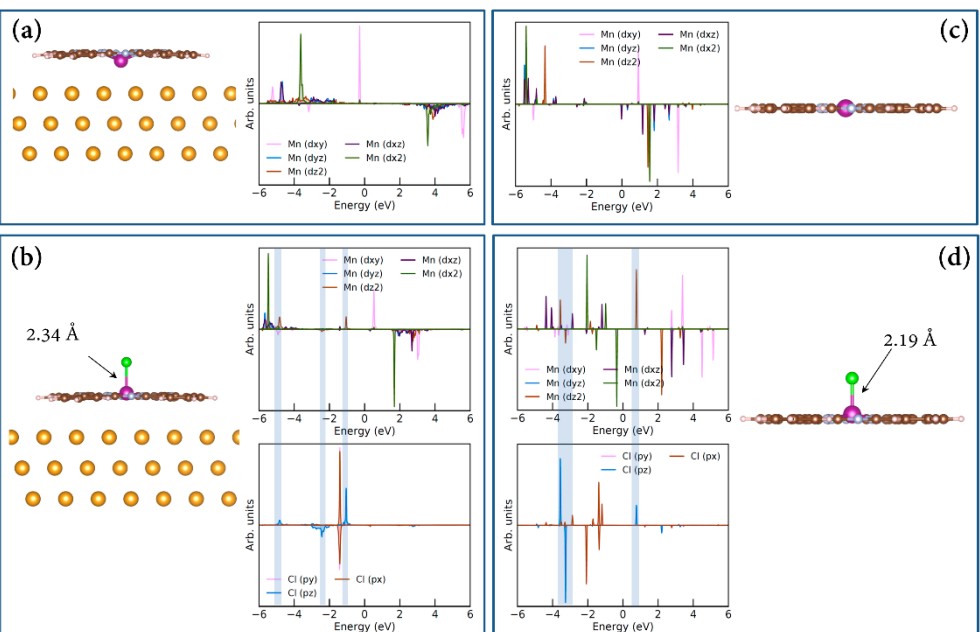

**Figure 4.** The results of DFT+$U$ calculations for MnPc and MnPcCl molecules. (**a**) Projected density of Mn d-states of MnPc adsorbed on Au(111), (**b**) projected density of Mn d-states and Cl p-states of MnPcCl adsorbed on Au(111), (**c**) projected density of Mn d-states of isolated MnPc, (**d**) projected density of Mn d-states and Cl p-states of isolated MnPcCl adsorbed on Au(111). Inset structures give side views of considered MnPc and MnPcCl molecules. For MnPcCl the overlap between Mn $d_{z^2}$ and Cl $p_z$ states is indicated. Fermi level was set to 0 eV.

## 3. Materials and Methods

Sample preparation: The Au(111) surfaces were prepared by repetitive $Ar^+$-ion sputtering and annealing. The cleanliness of the substrate was confirmed by X-ray photoelectron spectroscopy (XPS), where C1s, N1s, O1s, F1s, and Au4f signals were followed. XP spectra (Specs PHOIBOS 150) were acquired along the direction of the surface normal using monochromatized Al K$\alpha$ X-rays. MnPc, MnPcCl, FePc, CoPc, ZnPc, 2HPc, CuFPc, CoFPc (Sigma-Aldrich), and FeFPc (synthetized according to recipe in [77]) molecules were deposited on substrates kept at room temperature from self-made crucibles in the subsequent manner (vide infra). Crucible temperatures were adjusted to the molecule's sublimation point and rates were tuned using a quartz crystal microbalance (QCMB). The coverage, as well as the ratio between two molecular components, was checked by XPS and adjusted to account for close to a full monolayer (1 ML). To determine the total coverage, the overall C1s signal was compared to the substrate signal, namely Au4f, and correlated to the STM calibrated 1 ML C1s to Au4f ratio. The C1s to F1s signal ratio was used to determine component proportions, due to a fact that only one of the components would contain fluorine. The ratios were determined taking into account the analyzer transmission function together with sensitivity factors. Such prepared samples were then transported for further measurements without breaking the vacuum, using a portable UHV suitcase. During transport, which typically took 4 h, the pressure in the suitcase was closely monitored and did not exceed $1 \times 10^{-9}$ mbar.

X-ray absorption spectroscopy and X-ray magnetic circular dichroism (XAS/XMCD) measurements were conducted at the X-Treme beamline of the Swiss Light Source (SLS) at the Paul Scherrer Institute [78]. $L_{2,3}$ edges XA spectra were acquired in normal and grazing X-ray incidence. In normal XAS geometry, the incident X-ray beam and the vector of the external magnetic field were both parallel to the surface normal. In the grazing XAS geometry, the same formed a 70 degree angle with respect to the surface normal. All XAS/XMCD measurements were performed at the base temperature of 2.5 K, with or without an applied external magnetic field. For acquiring the spectra, the surface sensitive total electron yield (TEY) mode was used. Due to the large amount of data in the present investigation, a shorter acquisition time was chosen to resolve the peaks at a reasonable resolution, leading to higher noise levels (cf. Figure 2c,d blue vs. our previous report [5]).

Scanning tunneling microscopy (STM) experiments were performed with a low-temperature STM microscope (Omicron Nanotechnology GmbH with Nanonis SPM control system), as well as with a variable temperature Specs Aarhus 150 microscope using mechanically cut $Pt_{90}Ir_{10}$ tips. Tips were further prepared by in-situ Ar-sputtering or by pressing them into the clean Au(111) surface. In both systems, images were acquired in constant current mode and analyzed using the WSxM software [79].

Angle resolved photoelectron spectroscopy (ARPES) measurements were performed at the High-Resolution Angle Resolved Photoemission Spectroscopy (HRPES) endstation of the Surface/Interface Spectroscopy (SIS) beamline of the SLS. For the transfer of samples to the beamline, the vacuum suitcase was slightly modified to accommodate special sample holders that were used at the aforementioned beamline. Measurements were performed upon cooling down the samples to 10–15 K, the then lowest reachable temperature at the SIS beamline. The samples were first aligned to adjust the position of the $\Gamma$-point to the center of the scan, and energy–momentum dispersion was recorded using circularly polarized light with $E_{ph}$ = 76 eV in a sweeping mode (from $-0.7$ to 0.2 eV).

All the preparation and experimental steps were carried out under ultra-high vacuum conditions with the base pressure not exceeding $5 \times 10^{-10}$ mbar.

The first-principle DFT calculations were performed using the Vienna ab initio simulation code (VASP) [80–83]. The generalized gradient approximation (GGA) in the parametrization by Perdew, Burk, and Ernzerhof [84] combined with the projector augmented wave (PAW) method was used [85]. A cut-off energy of 400 eV and Gaussian smearing with a width of $\sigma$ = 0.025 eV for the occupation of the electronic levels were used. The on-site Coulomb interactions were added to the d-states of M using the DFT+$U$

scheme of Dudarev et al. [86], with $U$ = 4 eV at the orbitals of all metal centers considered here (M = Mn, Fe, Co, Ni, Cu). A k-point mesh of $3 \times 3 \times 1$ was employed for the integration of the first irreducible Brillouin zone [87]. MPc molecules were placed on the Au(111) surface constructed as a three-layer slab in a p($8 \times 8$) hexagonal supercell. The relaxation proceeded until the Hellmann–Feynman forces acting on all the atoms of the MPc molecules became smaller than 0.025 eV Å$^{-1}$. As the MPc molecules interacted weakly with the Au(111) surface [5,57], the surface was not relaxed. Spin-polarization was taken into account in all calculations, while dispersion interactions were treated using the D2 scheme of Grimme [88]. Graphical presentations were made using VASPKIT [89] and VESTA [90].

## 4. Conclusions

Supramolecular architecturing allows to create 2D checkerboard patterns with a wide range of coordination centers and specific electron and spin systems. Novel 2D spin arrays were prepared by 'supramolecular lego' after optimization of the fabrication and were characterized for their potential long-range order, among other magnetic properties. On the basis of the eight selected combinations of magnetic ions, the propensity of these systems to exhibit magnetic exchange coupling via Kondo and RKKY interaction involving the electronic states of the surface substrate was investigated. At 2.5 K, 2D-ordering was only observed after initial magnetization and for the Fe and Mn pairing. This observation is consistent with the partial occupancy of the d-states of the center-metal ions, as this suggests lower critical temperatures in most of the other cases or the absence of order due to complete Kondo screening. The particularly interesting case of Cl modified Mn combined with Fe suggests a pivotal role of the $d_{z^2}$ orbital and its occupancy for sustaining the RKKY interaction. DFT+U calculations provided deeper insight into the interaction of adsorbed Pc molecules on Au(111), providing a pre-requisite for the presence of RKKY interaction. Single layer systems, built by supra-molecular on-surface chemistry, can be locally tuned by out-of-plane ligation by scanning probes and globally as a cooperatively interacting ensemble. Depending on the interaction mechanisms, strength, and topology, these materials provide the basis for, for example, true single-layer two-dimensional (2D) quantum spin lattices, which are very difficult to realize otherwise. Thereby our work opens up new ways to create unprecedented architectures with atomic precision for future 2D materials and electron- and spin-based quantum technology.

**Supplementary Materials:** The following are available online at https://www.mdpi.com/article/10.3390/magnetochemistry7080119/s1, Figures S1–S7 as cited in the manuscript, discussion about ARPES experiments.

**Author Contributions:** M.B., O.P., J.G., J.N., J.D. and N.B. performed the XAS/XMCD experiments. M.B. performed ARPES measurements and analysis with help of N.C.P. and M.R., M.B., O.P., J.G., J.N., H.R., T.N., M.M., and S.F.M. were involved in the sample preparation and X-ray photoelectron spectroscopy analysis. O.P. and M.B. performed the STM measurements. Molecules were synthesized by X.L., S.D., and S.-X.L. XAS/XMCD data were analyzed by M.B. with help of J.D. STM analysis was performed by M.B., I.A.P. and N.V.S. performed ab initio calculations and analysis. M.B., S.D., I.A.P. and T.A.J. wrote the manuscript with input from all authors. T.A.J. and N.B. supervised the project. All authors have read and agreed to the published version of the manuscript.

**Funding:** This research was funded by the Swiss National Science Foundation (grant no. 200020-149713, 206021-121461, 206021-113149, 206021-144991, and 200020-153549) and the National Centre of Competence in Research "Nanoscience" (NCCR-Nano, project "Nanoscale Science"), Swiss Nanoscience Institute (SNI) (project nos. P1204 and P1203), Commission for Technology and Innovation (CTI) contract no. 16464.1 PFNM-NM, University of Basel. The computations were performed on resources provided by the Swedish National Infrastructure for Computing (SNIC) at the National Supercomputer Centre (NSC) at Linköping University, partially funded by the Swedish Research Council through grant agreement no. 2018-05973.

**Data Availability Statement:** Data is contained within the article or supplementary material.

**Acknowledgments:** We sincerely thank Rolf Schelldorfer for technical support as well as Mariah O'Doherty, Dorota Siewert, Foteini Ravani, Elzbieta Gradauskaite, and Andrew Hunter for support in the lab. We gratefully thank SIS and X-Treme beamline staff for their help.

**Conflicts of Interest:** The authors declare no conflict of interest. The funders had no role in the design of the study; in the collection, analyses, or interpretation of data; in the writing of the manuscript, or in the decision to publish the results.

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
