# Peer review of "Self-Assembly and Magnetic Order of Bi-Molecular 2D Spin Lattices of M(II,III) Phthalocyanines on Au(111)"

_magnetochemistry, doi:10.3390/magnetochemistry7080119_

Round 1
Reviewer 1 Report
The paper by M. Baljozovic et al. is a thorough study of bimolecular assemblies of phtalocyanines molecules on Au(111), both by ARPES and XMCD. This investigation of seven new systems is motivated by a previous finding by the same collaboration (Nat. Com. 2017) of a substrate driven antiferromagnetic coupling between spins of different metal centers in MnPc and FeFPc checkerboard lattice. The overall result is quite disapointing as none of the investigated systems display this property. However, I find that those 'no show' results deserve publication in order to help other researchers to better understand the spin coupling mechanisms in such systems.
To my opinion, the main new interesting measurements are in the supplementary informations, and I suggest to the authors to reorganize their manuscript to show them (maybe in a more concise manner, especially for XMCD) in the main manuscript. An important part of the present manuscript is dedicated to the MnPc-FeFPc which has already been published (both STM and XMCD), this instead could be in the SI. I have however an important remark on the data of Fig. 2. To my eyes, it is impossible to guess a positive XMCD signal at zero field at the Fe edge like in your previous Nature Com. Fig.2b, what completely changes your conclusion (and l.177-185 description) and also should raise a discussion on the reproducibility of such a measurement.
More technical remarks :
-it is not very clear how the calibration of the 1:1 stoechiometry has been done. You mention XPS, I suppose at the metal center energy, but how 2HPc is done ? Is XPS truely quantitative or have you done previous calibration
of the XPS signal with STM.
-it is of course of crucial importance to check that the bimolecular network arranges in a checkerboard pattern to observe an antiferromagnetic coupling with averaging technique. Have you checked all the systems by STM or only 2HPc-FeFPc (Fig. S5), from which I hardly recognize both molecules (maybe a sketch would be useful) ?
-can you precise the base pressure and time spent in your UHV suitcase before XMCD measurements ? Indeed, it is known that such metal center Pc can undergo some important changes in their spin state with adsorption of gaz impuritites.
-please indicate a cristallographic direction in the STM images so that people interested in simulating those systems can find enough details in your paper.
Reviewer 2 Report
My report is attached

Reviewer 3 Report
Comment.
The topic of the article covers an important issue of single layer low-dimensional materials. The authors develop on-surface supramolecular architecturing that has been employed to create surface supported two-dimensional binary spin arrays on atomically clean non-magnetic Au(111). The different checkerboards have been produced combining phthalocyanines containing metals of different oxidation and spin states. The correlation between the structure and the magnetic properties of these tunable systems has been established. The authors discussed the emergence of 2D Kondo magnetism from the spin-bearing components. The article is well written and well illustrated. In my opinion it can be published in the present version.
Round 2
Reviewer 1 Report
The authors have seriously answered to all the technical remarks (I note that the absence of signal at the Fe edge is a signal to noise issue, what was not so clear to me). I am still convinced that there are more new informations in the SI than in the main text but it is the choice of the authors to present their work in this form and to stick to it. All in all, it is an interesting paper using many different state of the art techniques to measure the magnetic properties of such 2D molecular networks. It can be published in its present form.